# Meaning-Making in Ecology Education: Analysis of Students' Multimodal Texts

Hanna Wanselin [1,*], Kristina Danielsson [2,3] and Susanne Wikman [1]

1 Department of Chemistry and Biomedical Sciences, Linnaeus University, 391 82 Kalmar, Sweden
2 Department of Swedish, Linnaeus University, 351 95 Växjö, Sweden
3 Department of Teaching and Learning, Stockholm University, 106 91 Stockholm, Sweden
* Correspondence: hanna.wanselin@lnu.se

**Abstract:** Teaching and learning in ecology depend on multimodality, involving semiotic resources such as visual representations, subject-specific symbols, and written and spoken language. Furthermore, the ecology field involves complex processes and relationships, presenting student challenges. However, more research has yet to investigate how students design multimodal texts to represent complex biological processes. For a holistic understanding of ecology, it is crucial to understand different complex processes, such as the matter cycle, energy flow, decomposition, and their relations. Therefore, this study aims to, through multimodal text analysis based on systemic functional linguistics (SFL), identify how secondary students collectively present and combine such processes and how they position themselves through their textual choices. Results indicate that representing biological processes comprises several challenges for students. One way in which this is shown is the unclear use and meaning of arrows. Thereto, the students include various aspects uncommon in the field of ecology, for example, symbols inspired by comic books, values, and the role of humans, thereby relating ecosystems to their interests and everyday life. Implications for teaching are discussed, for instance, the importance of supporting students in terms of scientific content and how to represent it, which can be conducted through text discussions.

**Keywords:** multimodal texts; systemic functional linguistics; ecology education

## 1. Introduction

Ecology has a prominent role in curricula across all levels of schooling, from elementary to upper secondary school [1–4]. Yet it has been identified as a challenging content area for students since its study involves making meaning about a number of complex biological processes and their connections [5–8]. The expectation is that students acquire a holistic understanding of ecosystems, living organisms, and their mutual relationships in a specific environment, to take a stand on ecological issues, such as climate change, energy supply, and resource utilisation [3].

In teaching resources, ecology is presented predominantly in ways that utilise a variety of semiotic modes—such as images and written words. Students are also expected to draw on a range of multimodal resources when expressing their ideas about ecology. However, to date, few studies have been based on multimodal analysis of students' texts as a way of discerning how they interpret, combine, and make meaning of scientific processes in biology (but see [9]) and how they position themselves in relation to the subject or discipline through their choices. Herein lies this study's aim—building on detailed analyses of students' multimodal texts collected in three Swedish secondary classrooms—to investigate how secondary students express complex biological processes through different resources in multimodal texts. Further, with this analysis as a basis, we can gain insights about how the students, through their texts, position themselves in relation to the subject content.

## 1.1. Previous Research

In their ecology studies, students are expected to make sense of multiple-linked subsystems with processes and relationships that occur at macroscopic and microscopic levels [7]. The challenge here is that whilst the macroscopic level is observable, the microscopic level is not. Previous research in ecology tends to concern itself with matters such as students' misconceptions (see review [10]). The relationship between energy and matter, predator-prey interactions, and decomposition have been identified as examples of content covered in ecology that presents specific challenges (see [5,10–14]). Other challenges in ecology education relate to the use of visual resources, such as diagrams, involving unclear starting points, reading pathways, and the meaning of arrows [12,13]. Arrows are essential in many visual resources in science. A reason for students' challenges in interpreting and using them may be their unclear and unarticulated function and meaning [15], involving, for example, direction, pathways, movement, temporality, and causality [16,17].

Both in classroom interaction and textbooks, biological processes, such as photosynthesis and circulation of matter, are often presented separately without reference to their relationship [18,19]. Research in ecology has tended to fall into the same trap, focusing on details of isolated processes and phenomena [10]. Examples are photosynthesis [20–22], decomposition [14], and food webs and chains [12,23,24]. Yet, for students to form a more comprehensive view of ecosystems and to be able to understand their role in acting for sustainable development, an awareness of relationships between such complex biological processes is needed [5]. However, earlier research shows that students find it challenging to merge biological sub-processes into a whole, for example, processes in the human circulatory system [6], aquatic systems [25], and ecosystems [5,7,8]. Similar to many other scientific processes, biological processes are often complex and impossible to observe. In the case of ecology and, for example, the transfer of matter through an ecosystem, excursions in nature cannot support students to discover the processes, as is the case in other areas, such as national park interpretation [26]. Students might notice parts of the ecosystem, for example, worms contributing to decomposition, while the transfer of matter in the ecosystem cannot be observed.

## 1.2. Multimodality in Science Education

Seeing as many scientific phenomena are often too small, large, abstract, or complex to be perceived directly through our senses, the science discourse used to describe them tends to be highly multimodal, utilising resources in several semiotic modes, such as image, writing, and gestures [9,13,27]. Accordingly, a variety of representations symbolising a concept, phenomenon, or process, are common in science [28], e.g., the diagram representing a food web or a food pyramid in Appendix A. In recent decades, there has been increasing interest in the analysis of how visual representations are employed as teaching resources [29–31], and specifically, how students create representations through drawings [9,32,33]. However, few studies have focused specifically on ecology (see [5,23,34–37]). Out of these, even fewer concern issues connected to multimodality, for example, students' multimodal awareness when interacting with websites [36] and teachers' multimodal communication in classrooms [37], though analysis of students' multimodal texts is rare. What research has been undertaken (see, for instance, [38]), suggests that as students create multimodal texts to describe and explain scientific phenomena, this has considerable potential to enhance their meaning-making. Furthermore, such texts can also be used by teachers to assess students' views of scientific phenomena [13,39,40], another aspect that this study seeks to explore.

## 1.3. Aims

The aim of this study is to investigate how secondary students through different resources express and combine complex biological processes to form a multimodal whole in their texts and how they through their choices take a stance related to the content. This is conducted by analysing a number of group texts created by secondary students. Our research questions are:

- What aspects of biological processes are expressed in student texts, and through what resources?
- How do students position themselves, for instance, regarding subject content, through their texts?

The analysis was performed using systemic functional linguistics, SFL [41], which is useful for analysing complex relations in texts. It enables parallel analysis of different modes, such as writing and image, concerning which content is expressed and how. Hence, such analysis can provide indications of students' ideas and how they position themselves (see methods section), thereby highlighting their varying views of ecology.

### 1.4. Theoretical Perspective
Social Semiotics and Multimodality

To elucidate students' meaning-making through multimodal texts where written and drawn elements are combined with symbols, this study is based on social semiotics [42,43], including perspectives on multimodality [9,44]. From a social semiotic point of view, form, and function are inseparable; thus, how the content is presented strongly influences the actual content expressed [45]. From this perspective, sign-making, meaning-making, and learning are closely related [46]. A key term in social semiotics is mode, i.e., a culturally developed system of semiotic resources, for example, image, speech, or gesture [44]. Each mode has its specific affordance [17] or potential for meaning-making. For instance, images are described as being better suited for expressing spatial arrangements than writing, while writing is better suited for reasoning about cause and consequence [9,46]. Therefore, human interaction is inherently multimodal, involving several modes forming a whole [46].

In line with Jewitt [45] and Kress and Selander [47], we view the student texts as students' choices regarding what to represent, and how and where their texts can be seen as a redesign of the teacher's and teaching resources' designs for learning. Apart from the content knowledge they might possess, potential differences between student texts can be seen as expressions of their different earlier experiences and/or interests [9]. In this study, we have focused on one product of the students' meaning-making process, namely the produced texts, and we interpret their choice of signs as evidence of their meaning-making process [47]. However, taken as such, the texts are seen as products of the students' meaning-making process in a specific social setting, rather than corresponding to their actual knowledge or cognitive capacities.

## 2. Materials and Methods
### 2.1. Data Collection and Context

The data consists of multimodal student texts created by students in three Grade 7 biology classes at two private schools located in a small city in Sweden. The schools were chosen based on a convenience selection: they are located within a reasonable geographic distance, had interested, and educated teachers, and the ecology education was carried out during the time for the planned data collection. Both schools are situated in the same area, with students from a variety of socioeconomic backgrounds. Each of the three participating classes had 24–25 students, between 13 and 14 years of age. Over a two-month period, the first author attended the classes to observe and video-record the classroom activities. The teaching was planned and carried out by the teachers. The research team provided the teachers with a suggested assignment for the students in which they were asked to create texts depicting the functions and relations of a food web. In addition to this task, the assignment contained several questions to support the students in developing their answers (see Appendix A). They were given two visual resources from their textbook, depicting a food web and a food pyramid. The students were placed in groups of four by their teachers and created texts on paper sheets (A1 format), using coloured pencils.

A total of 17 texts were generated in the three classes. The students were given up to one and a half hours to complete the assignment. In addition to the visual resources provided to them through the assignment, they also had access to their textbooks. The

teachers interacted with the students as was his or her normal practice. Although all 17 texts were analysed for the present study, five texts were selected as illustrative examples for discussion in the results section below. These five texts show varying ways of representing biological processes. The study follows the Swedish Research Council's [48] ethical principles, such as consent (written consent from teachers, students, and caregivers), the right to withdraw consent, and guaranteed anonymity.

### 2.2. Analytical Procedures

To undertake a systematic analysis of student texts, we utilised a framework (see [49]), based on Systemic Functional Linguistics, SFL [41]. The framework applies aspects of Systemic Functional Grammar, SFG, combined with several SFL applications for analysing visual representations [17,50] and relations between written and drawn elements [51]. With its origin in social semiotics [42], in SFL, the language's function is emphasised where each text simultaneously realises three different meanings through three metafunctions: ideational, textual, and interpersonal [41,52]. The ideational metafunction concerns the content expressed, whilst the textual metafunction relates to the organisation of the text. Both of these metafunctions are connected to our first research question, concerning aspects of biological processes that the students express through their multimodal texts, and the choice of, and arrangement of, the resources used to express the content. In the analysis, lines, and arrows were integrated with drawn elements, whereas chemical formulae were integrated with writing. The interpersonal metafunction concerns relations, for instance, how the text producer relates to a potential reader and the content [41]. In this study, we relate the interpersonal metafunction to how students position themselves concerning the subject (our second research question). The text analyses were intersubjectively validated through independent and joint analyses by the first and second author. Earlier research in science education has predominantly focused on the ideational metafunction [31,51,53,54] whereas we base our analyses on all three metafunctions. The analytical framework is presented in detail in Wanselin, Danielsson, and Wikman [49]. Specific aspects of the framework employed in this study are given in Table 1 and they are presented in the following sections.

**Table 1.** The analytical framework used in this study (based on [49]).

| Ideational Metafunction | Textual Metafunction | Interpersonal Metafunction |
|---|---|---|
| transitivity analysis -written/drawn elements (processes, participants, and circumstances) -narrative/conceptual function of drawn elements | organisation of text choices of specific resources (drawn/written elements, subject-specific symbols) | lexical choices and drawn elements in relation to science discourse |
| | relative size and scale | explicit/implicit values |
| relationship written/drawn elements (redundant, complementing, elaborating, contrasting) | | |

### 2.2.1. Ideational Metafunction

We performed a transitivity analysis of the students' written and drawn elements, including whether the visual resources had a narrative or conceptual function. The relationships between drawn and written elements were then analysed in terms of how they were related, and their respective levels of generality. These different steps are further explained in the following.

The transitivity analysis identified processes, participants, and circumstances in written and drawn elements. The first step was to identify processes (note: the term 'process' refers to linguistic processes, whereas biological/scientific processes are referred to in terms of, e.g., 'biological process', and not just 'process'). In verbal texts (writing, speech), differ-

ent verb types correspond to different process types (examples within brackets): material ('moves'), mental ('feel'), verbal ('say'), and relational processes ('is'). Halliday [41] includes existential and behavioral processes. In line with commonplace applications of SFL for Scandinavian languages such as Swedish [55], we incorporate existential processes with relational processes and behavioral processes with material ones. The processes involve participants with different roles depending on the type of process—an actor in a material process, a senser in a mental process, and a sayer in a verbal process. Relational processes relate participants to each other. Participants are realised by nominal groups, such as 'food web', 'fish', and 'she'. Circumstances, finally, are expressed by adverbial groups and prepositional phrases, giving information related to questions such as 'how', 'when', or 'why'. In line with Kress and van Leeuwen [17], drawn elements (including vectors, such as arrows, and drawn lines with a direction, such as sun rays) that indicate material processes, such as a movement, were categorised as narrative. Drawn elements indicating relational processes, were categorised as conceptual—for instance, an image highlighting the relation between a whole and its parts. The relationship between drawn and written elements was analysed in terms of redundancy (i.e., same content in both modes), extension (information in one mode complements the other), elaboration (content in one mode act as clarification or exemplification), and contrast (content in one mode is contradicting the content given in the other mode) (cf. [31,51,56]).

### 2.2.2. Textual Metafunction

Our analysis considered how the text was organised in terms of the overall layout, choice of modes, and how different resources are combined and relate to one another, which, taken together, can indicate a potential reading order (cf. [56]). Spatial aspects, such as the relative size and scale of drawn elements, were also included in the analysis concerning this metafunction (cf. [50]). Relative size considers whether depicted objects are of similar size, while relative scale considers whether the size, position, and spacing of objects are depicted realistically and proportionally.

### 2.2.3. Big Ideas Identified through Analysis of the Ideational and Textual Metafunctions

Once all student texts had been analysed, it was evident that they all contained aspects of content that relate directly to three so-called 'big ideas' [57–59]. A big idea is a phenomenon or concept considered most important for students to learn in a certain field [59], and they often constitute part of the core content of school curricula. In the student texts, the big ideas of photosynthesis as the foundation of life, energy transformation, and matter circulation (cf. [58,60,61]) were identified. In terms of the ideational metafunction, written or drawn participants, processes, and circumstances, and combinations of them, can be related to these big ideas. For example, an image of a sun and plant combined with chemical formulae such as $H_2O$ and $CO_2$ indicate the photosynthesis process. Similarly, concerning the textual metafunction, the placement of elements in relation to each other can also relate to big ideas. For example, where species were drawn in such a way as to imply that energy flows or that matter circulate.

Seeing as photosynthesis and energy transformation relate to the energy flow in an ecosystem, overlaps between these big ideas in the student texts are expected. There are also overlaps between energy flows and matter circulation since organisms' feeding patterns involve both energy and matter. Thus, textual resources may relate to more than one big idea—for example, where species are shown as being connected, this can indicate both energy flow and matter circulation.

### 2.2.4. Interpersonal Metafunction

The students' choice of lexicogrammatical and visual resources can reveal how they position themselves in relation to the subject [62]. One way in which they can position themselves is as being knowledgeable in the field (cf. student positions such as 'nerd' and 'fact-oriented' in Lyng [63] and Løvland [64]), is to use disciplinary-specific terminology or

images (choices characterised as 'formal' by Christidou and colleagues [33]). In contrast, through the use of everyday words, or even images similar to those in comic books, students may be seeking to distance themselves from the discipline, or even appear funny (cf. 'entertainer' positioning in Løvland [64]). Finally, students' choices of written and drawn elements can be analysed with respect to explicit and implicit values, for example, the use of words such as 'unimportant' or 'good,' or visual resources expressing values or norms where aspects such as right or wrong come to the fore [56].

## 3. Results

To address the first research question, 'What aspects of biological processes are expressed in the student texts, and through what resources?', the results section is structured around four sections dealing with content that was identified through the analysis. The first three sections deal with the three big ideas that were identified through the analysis of the ideational and the textual metafunction, namely photosynthesis, energy flow, and matter circulation. These are all directly connected to the subject content that had been focused on during teaching and learning activities as well as the assessment. The fourth section deals with the role of humans, which was also identified as a recurring theme through the ideational analysis. The relationship between drawn and written elements, which is part of the ideational metafunction, is presented for all big ideas and the role of humans taken together since such relationships are not specific to any of the big ideas or the role of humans. Similarly, the results of analysis concerning the interpersonal metafunction, which relate to the second research question: 'How do the students position themselves in relation to the subject content through their texts?', are also presented for the big ideas and the role of humans taken together. As mentioned, the results section is based on the analysis of all 17 student texts, though the five texts shown in Figures 1–5 are used as examples. The entire data set shows differences concerning both scientific complexity and the use of different semiotic resources, which is evident from the chosen examples. In the following, comments such as 'all texts' or 'few texts' refer to the whole data set.

Figures 1–5 shows the texts chosen as examples to illustrate the results. These texts have different characteristics. In the original texts, writing was made by hand. In Figures 1–5, the writing has been translated from Swedish into English. Group A's text (Figure 1) involves four separate sections, all taken from the assignment and the students' textbook (Appendix A). This is the only text without arrows. Both group B's and D's texts represent complex ways of representing biological processes (Figures 2 and 4) through the inclusion of decomposers, chemical formulae, and scientific terms. In contrast, group C's text represents a simple food chain structure involving start- and endpoints, lacking decomposers (Figure 3). Furthermore, this text emphasises the water cycle. Group E's text shows a 'concept map design', emphasising humans and human relations (Figure 5).

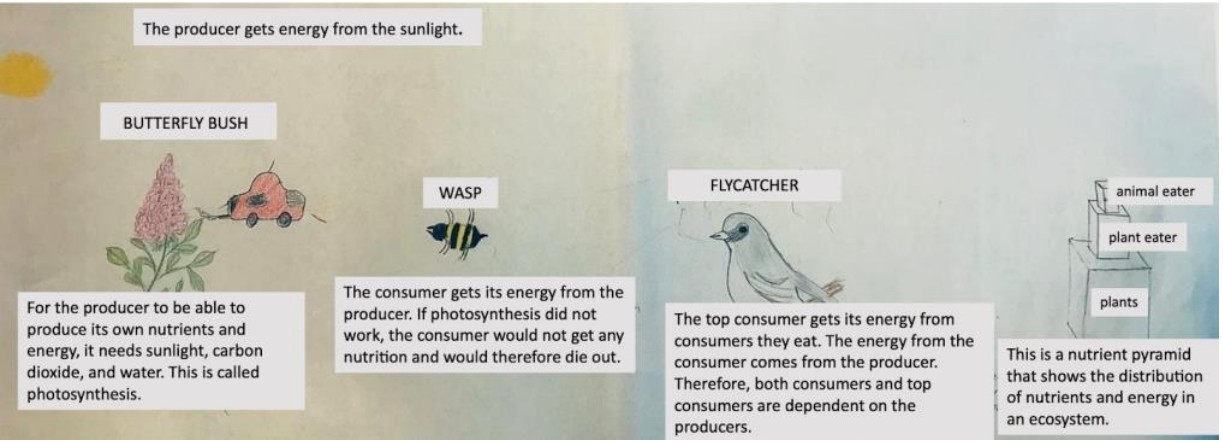

**Figure 1.** Group A's text.

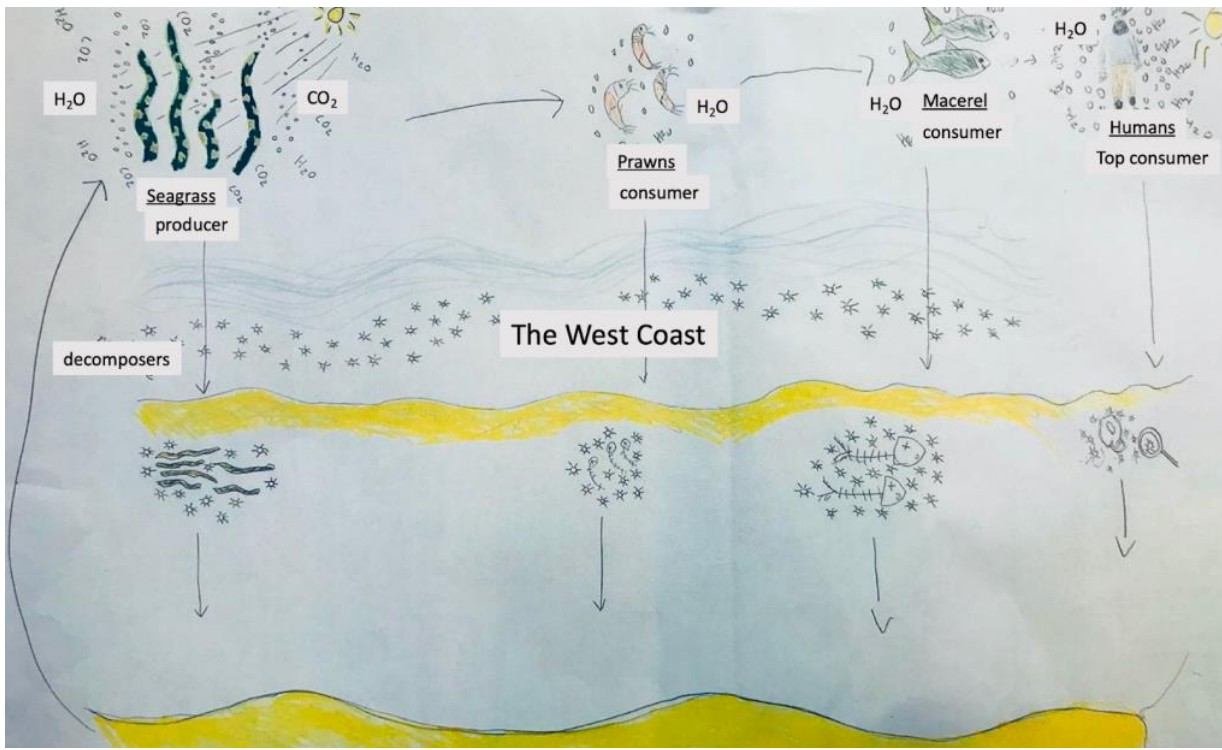

**Figure 2.** Group B's text.

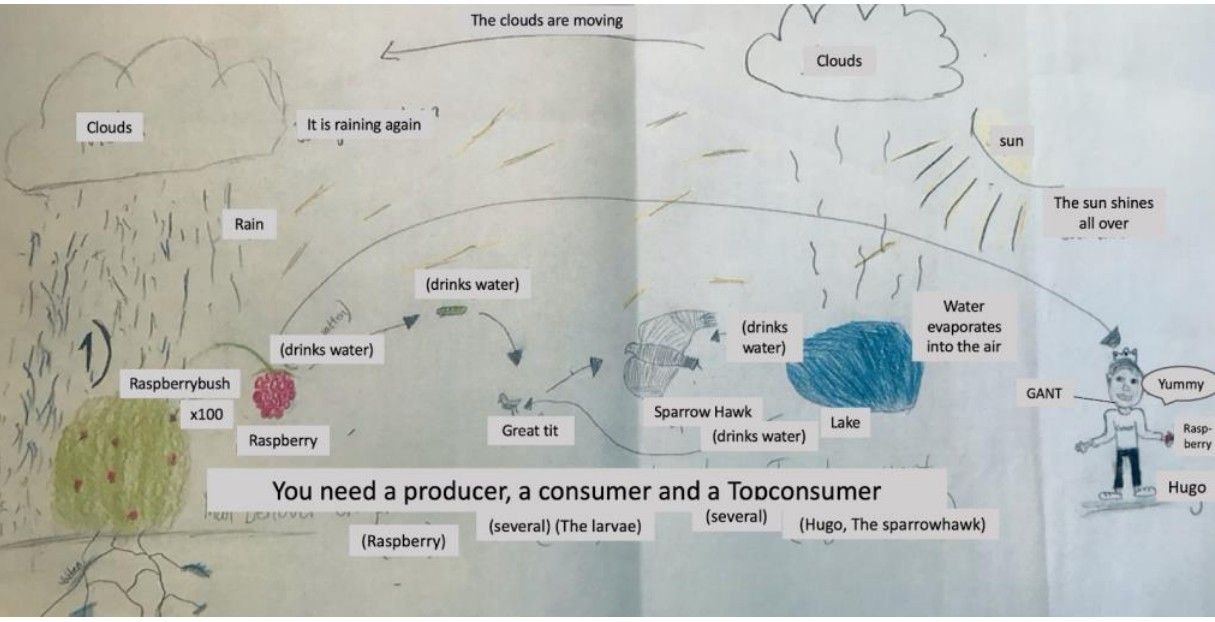

**Figure 3.** Group C's text.

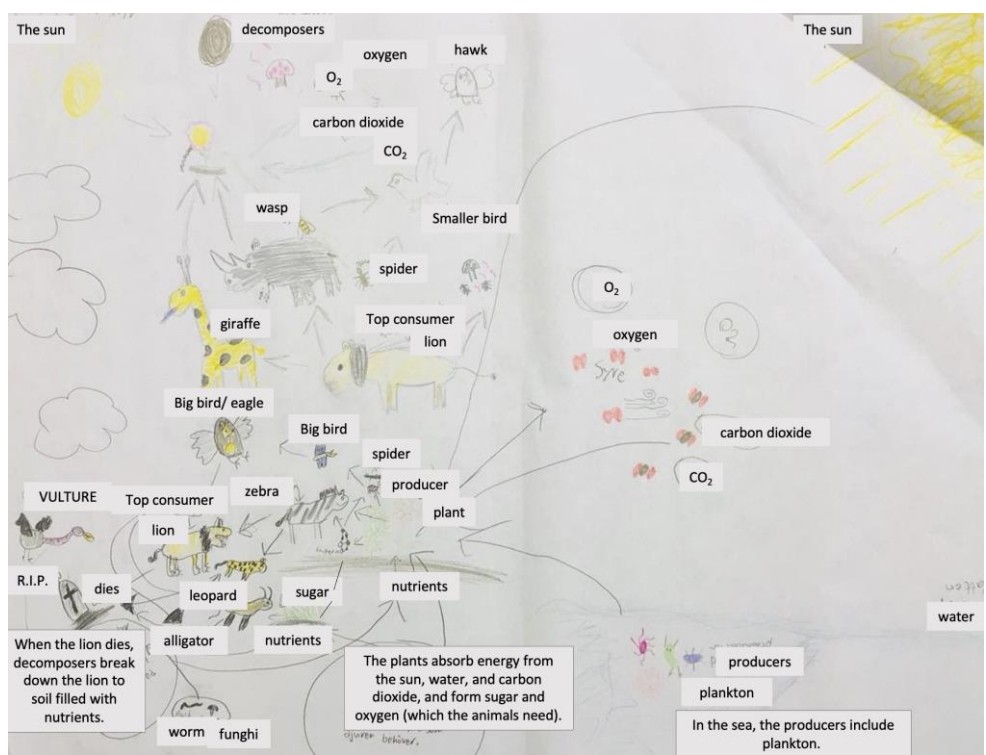

**Figure 4.** Group D's text.

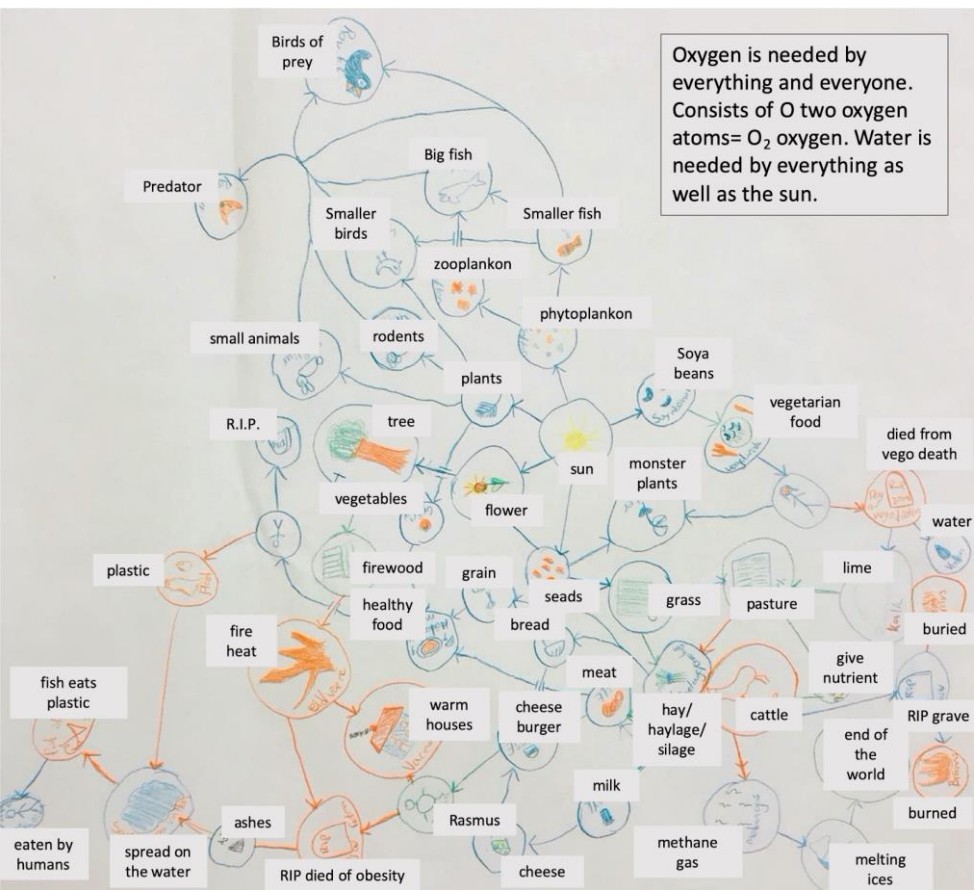

**Figure 5.** Group E's text.

### 3.1. Photosynthesis

3.1.1. Ideational Metafunction

The transitivity analysis revealed that in all but one of the student texts, the students use writing, images, or a combination of both to represent the elements that are commonly identified as central parts of photosynthesis (i.e., plants, the sun, and water). Typically, this is conducted by including an image of the sun as a participant in a material process, with vectors (i.e., lines or arrows pointing from the sun) depicting sunrays. In 13 of the 17 texts, arrows linking plants to representations of oxygen, carbon dioxide, and water indicate an uptake, release, or movement. The representations employed are images, words, chemical formulae, depictions of molecules, or different combinations of these (see, e.g., Figure 4). The written texts include words such as 'producer', 'energy', 'water', and 'glucose molecules' (Figures 1–5), or images of the sun or plants (e.g., Figure 2). In a few texts, the term 'photosynthesis' is included even though the term was not mentioned in the assignment. Regarding process types connected to photosynthesis, material processes are common in writing and image, thereby focusing on photosynthesis as a scientific process. Examples are the words 'get', 'form', and 'take in', and vectors indicating material processes. Written relational processes or drawn elements of parts of photosynthesis without arrows (Figure 1) result in a conceptual function (cf. [17]) identifying parts of photosynthesis. An example is the description of what a producer needs, followed by 'This is called photosynthesis' in Figure 1. In some cases, circumstances related to place, such as 'from sunlight' (not in examples) and cause, 'for the producer to be able to produce . . . ' (Figure 1), indicates photosynthesis. Circumstances are often realised in drawn elements, for example, by showing where something emanates from. An example is Group D's text (Figure 4), where carbon dioxide from the air is taken up by plants.

3.1.2. Textual Metafunction

Photosynthesis was also identified through analysis of the textual metafunction. In Figures 2–4, for example, photosynthesis is depicted as the foundation for the ecosystem. Regarding overall layout, vectors from the sun to a producer (a plant) indicate a starting point for the ecosystem (e.g., Figures 2–5). Occasionally, water is linked to the producer by vectors (not shown in the examples). In one text (Figure 4), drawn molecular models (oxygen atoms in red and carbon atoms in black) are included. The sun is often displayed in a prominent position through its large size and/or central placement in the text (e.g., Figure 5). Predominantly, drawn elements related to photosynthesis are of similar size (e.g., plant and molecules in Figure 4). In that way, the focus appears to be on the system rather than, for instance, individual species. Elsewhere, the relative scale of objects is more proportional, e.g., the size difference of gas bubbles and seagrass in Figure 3.

### 3.2. Energy Flow

3.2.1. Ideational Metafunction

As mentioned, since the big ideas concerning photosynthesis and energy relate to energy flow in an ecosystem, overlap between these ideas is anticipated. All texts include drawn and written elements connected to energy, typically through the inclusion of organisms in a food chain or web structure and a sun (e.g., Figures 2–5). In writing, several student groups use the term 'energy', and others combine the words' energy and nutrition 'For the producer to be able to produce its own nutrients and energy . . . ' (Figure 1). As with photosynthesis, material processes are referenced through words 'come' and 'get' (e.g., Figure 1) or images (e.g., through vectors) concerning energy flow. Participants involved in these material processes are, for example, words such as 'energy' and 'top consumer', and students link energy flow to organisms' feeding patterns, for example: 'The top consumer gets its energy from the consumers they eat' (Figure 1). Arrows imply material processes, for instance, that a producer is eaten by a consumer (Figures 2–5), resulting in a narrative function (cf. [17]) in all but one of the student texts. Significantly, however, labels seldom clarify the meaning of arrows, leaving the interpretation to the reader—examples can be

seen in Figures 2 and 4, where the arrows can be interpreted as representing either matter transfer and/or energy flow. Occasionally, the arrows are drawn in an inconsistent manner, at times resulting in contradictive directions (not in the examples). Furthermore, in some texts, we could note that the students had erased the heads of some arrows. Circumstances connected to energy flow are, for example, 'from the producer' (e.g., Figure 1), relating to place, while chained arrows imply events occurring in a specific time order (e.g., Figures 2–5).

### 3.2.2. Textual Metafunction

As mentioned, the sun often acts as a starting point in the students' depiction of the ecosystem (e.g., Figure 5). As such, the inclusion of the sun can also be interpreted as an indication of the big idea concerning energy. Furthermore, top predators, or a human (e.g., Figure 5), act as endpoints in the ecosystem. At times, drawn elements are disproportionate and of equal size, for example, in Figure 2, where a human is the same size as the prawns and the objects are placed at equal distances, hence, the focus appears to be on the system rather than for example on specific species.

### 3.3. *Matter Circulates*
### 3.3.1. Ideational Metafunction

The student texts include the transfer of matter through organisms' feeding patterns, decomposition, and cyclical tendencies. As mentioned, depicted organisms are connected by arrows in all texts except one, which can indicate matter transfer or circulation within a system. Furthermore, over half of the student texts include decomposers in drawn elements, such as worms and fungi (e.g., Figure 4) or the term 'decomposers'. In captions, some groups use the abstract terms 'nutrients' and 'matter' which are central to the circulation of matter in an ecological system. Material processes connected to the circulation of matter, such as 'die' and 'break down', are often given in writing (Figure 4) at times combined with participants such as 'producer' and 'decomposer'. In drawn elements, material processes are realised through arrows in all texts except one (Figures 2–5). The possible meanings that can be ascribed to these arrows are: 'which organism is eaten by another', 'something dies', or is being 'decomposed' (e.g., Figure 2). Arrows (straight or curved) point from top consumers to decomposers or from decomposers to producers. Curved arrows between different species imply matter transfer in the system. Some circumstances that can be related to the circulation of matter concern time 'When the lion dies . . . ', while others relate to where something emanates from, e.g., the placement of drawn elements, e.g., the leopard obtains something (e.g., matter) from the zebra (Figure 4).

### 3.3.2. Textual Metafunction

The overall structure of the texts and the organism's placement in food chains or webs within either a chained or circular structure relates to matter transfer and circulation throughout ecosystems. In these cases, objects are linked by arrows, with plants as starting points and top predators as ending points (e.g., Figure 3). Group E's text (Figure 5) consists of several circular elements of similar size connected with arrows in a concept map design. Its layout forms a complex visual resource, very different from the other texts and resources commonly used in biology textbooks.

### 3.4. *Role of Humans*
### Ideational Metafunction

Seeing as more than half of the texts include depictions of humans (e.g., Figures 2, 3, and 5), the role of humans in ecosystems has been included in the analysis. However, the results connected to the ideational metafunction alone are presented in the following since the students' textual choices did not contribute to expressing this big idea. Transitivity analyses of drawn and written elements present a picture of humans as being one species amongst others and as a top consumer in the food chain/web (e.g., Figures 2, 3, and 5).

This role is realised through implicit relational processes, for example as in Figure 2, where 'humans' are 'top consumers'. In some texts, humans are also linked to photosynthesis, for example, as a source of carbon dioxide. Drawn elements including humans also realise material processes, for instance, 'Rasmus', who eats a hamburger (Figure 5). In one text (Figure 3), humans ('Hugo') are represented as consumers (of raspberries) and, implicitly, as goods (fashion label 'GANT'), implying a broader human impact on ecological systems. The human impact on nature is also realised by including how eating meat and breeding cattle leads to 'melting ices (sic)' and the 'end of the world' (Figure 5). Occasionally, the processes related to other circumstances, such as time, as in 'when humans eat . . . ' and a connection to decomposition 'when we died and became soil' (not shown in examples).

### 3.5. Relationship between Drawn and Written Elements

The relationship between drawn and written elements is considered for all three big ideas and the role of humans taken together. In most instances, drawn elements realise more processes, participants, and circumstances than written elements. One such example is shown in Figure 4, where the written captions relate mainly to decomposition and photosynthesis, without mentioning the food web shown through the drawing. Figure 2 depicts a food chain and food pyramid in combination—seagrass (producers), prawns (consumers), and humans (top consumers) represent the food chain, and the diminishing number of each species in the drawings represents the food pyramid. Interestingly, this is a connection that is not made in the teaching resource made available to the students. Whereas, in some instances, drawn and written elements are taken together to enhance the amount of information provided in the texts (e.g., Figure 4), in others, they realise several similar processes, participants, and circumstances, leading to redundancy (e.g., evaporating water in Figure 3). Examples of elaboration are when images at a species level are combined with labels describing organisms at a more general and abstract level (e.g., the label 'producer' next to an image of a sunflower). However, in some texts, labels on both a species and general level are connected to a drawn element of a species, for example, the labels 'prawn' and 'consumer' are connected to drawn prawns in Figure 2. No contrasting relations between information given in different modes were noted in the texts.

### 3.6. Interpersonal Metafunction

Concerning the interpersonal metafunction, the analysis suggests that the students, to a great extent, seem to seek to position themselves as being knowledgeable in ecology. This is achieved in various ways—such as the use of subject-specific terminology ('producer') to present scientific facts, and the inclusion of drawn molecular models (e.g., red oxygen atoms and black carbon atoms in Figure 4) and chemical formulae (e.g., '$CO_2$' and '$H_2O$' in Figure 2). Such choices correspond to a high level of formality [32]. Group A's choice to include the same species found in the textual resources provided to them, can be interpreted as an attempt to position themselves as 'good' students, sticking closely to the brief of the assignment. Yet, many of the students' drawn elements include concrete, naturalistic, and detailed elements. Taken together with the more personal, everyday language, this results in a lower level of formality with a corresponding lessening in the degree of scientific positioning. The students also express creativity through their choices, as illustrated in Figure 5, where the students produced a text in a 'mind-map' design, which incorporated symbols from their everyday lives. To varying degrees, all student texts displayed a level of inventiveness. For example, regarding the choice of organisms and habitat, the textual resources they were provided with (see Appendix A) only reference terrestrial organisms, whereas the students' texts included both terrestrial and aquatic organisms. In addition, the students inject a degree of humour through the inclusion of symbolism inspired by comic books (Figures 2–5). Such symbols are found in half of the texts, and examples are crosses over the eyes (Figure 3), a gravestone with the inscription 'RIP' (rest in peace) (Figure 4), an organism lying with feet upwards (not in examples), all indicating dead organisms. Students also include strains of fantasy, for instance, by describing that people can be killed

by a 'monster plant' (Figure 5). Such symbols and lexical choices lead to lower formality, suggesting that the students relate science to broader cultural domains and everyday life.

The choices made in some texts position the students as environmentally conscious through explicit and implicit values. The inclusion of humans in more than half of the student texts is particularly noteworthy. When including humans, the students reveal a critical stance on humans' impact on an ecosystem (also see the role of humans, above). Examples are humans as consumers of goods and Group E's (Figure 5) description of cattle breeding and meat-eating leading to increased methane gas levels. Other examples are written comments such as 'the end of the world' and the labels 'died from vego death' and 'died from obesity'. An example of the assertion of more implicit values is a human with a crown and expensive clothing brands in Figure 3.

## 4. Discussion

Even though the creation of multimodal texts in science has received attention in previous research, to our knowledge, few—if any—studies have been performed in biology classrooms dealing with complex processes such as the flow of energy, circulation of matter, and decomposition from a multimodal perspective. By investigating students' multimodal texts, we aimed to investigate how secondary students express and combine such complex biological processes through different resources and how they position themselves through their textual choices. In the following, we highlight some important findings, structured around the research questions and metafunctions, followed by implications.

### 4.1. Ideational Metafunction

Predominantly, material processes dominate the texts, both in terms of writing and images. This indicates a focus on the scientific processes, for example, on photosynthesis as a whole, rather than on the sub-processes or parts involved (e.g., light energy, water, carbon dioxide, glucose). If the focus had been on classifications, relational processes would have been expected, too (cf. [61]). In this study, different possible meanings of material processes and arrows connecting participants in processes are: 'one organism is eaten by another', 'something dies', 'something becomes decomposed', or 'substances are absorbed or released'. Occasionally, arrows and arrowheads have been erased and used inconsistently, implying insecurity to both the subject content and how to use arrows in line with the conventions of science. This result aligns with previous research indicating that interpretation and use of arrows are challenging for students [12,13,23,24]. Even though the students connect to the big ideas concerning matter and energy, the unclear meaning of arrows gives the impression that food chains and food webs concern who eats whom rather than energy flow or the cycle of matter. Participants connected to processes (mainly through arrows) are often drawn organisms, predominantly presented at a species level. This is similar to how ecosystems are commonly represented visually in textbooks (shown in Appendix A). However, when the students combined drawn species with labels, the writing on the label was often given on a more general level than that of the drawn element. This can be related to the affordance of an image as a semiotic mode, leading to a challenge to depict abstract participants, such as producers. Students' inclusion of scientific terminology in writing ('producer') also indicates an awareness that food chains and webs involve more than 'which species eat which' (cf. [12]), which would be a more everyday idea of scientific phenomena. Furthermore, by including scientific terminology rather than everyday words, the students show that visual resources, such as food webs, are generalised scientific explanatory models where specific species can be replaced by another species at the same trophic level. Decomposers' role in ecosystems is crucial for understanding how matter circulates, and that matter can neither be created nor destroyed. In the student texts, predominantly, decomposers are linked to top consumers when in reality, all dead organisms are decomposed. This indicates that understanding the role of decomposers might be challenging for these students, which is in accordance with earlier research [11,14].

The transitivity analysis also revealed that the students included humans in the ecosystems, at times highlighting negative aspects, such as humankind's contribution to global warming. To achieve sustainable development [65], society needs citizens who can make conscious and sustainable decisions. Thus, it is central to educate students toward understanding their own impact on society and nature. The inclusion of humans indicates such awareness. However, neither the teaching nor the assignment encouraged the students to include human impact in the ecosystems.

The inclusion of humans in the student texts can also be a way to connect ecology to everyday life. This was mainly carried out through everyday language and with concrete, at times humoristic images (also see below). It is well known that teaching based on students' experiences and interests may positively affect their attitudes toward school science, which may increase possibilities for students' meaning-making [66]. One student text largely relates to humans and for example, what humans eat and their dietary requirements (Figure 5). The focus in this regard is encouraging, particularly in light of earlier research indicating that students often do not connect the food they eat with food webs and ecosystems [67]. Group E's (Figure 5) inclusion of humans in ecosystems indicates that they have achieved a degree of a broader understanding of the role that humans play in pursuing a sustainable society.

### 4.2. Textual Metafunction

Research in the field of ecology indicates that students are generally aware of details of photosynthesis (such as chemical formulae), without being aware of its fundamental role in ecosystems [22]. In contrast, most students in this study seem aware of photosynthesis's importance for an ecosystem, and that photosynthesis is the base for life on earth, even though photosynthesis was not emphasised in the assignment. Furthermore, photosynthesis is represented as the starting point for the whole ecosystem by placing images of the sun and plants in combination with oxygen, carbon dioxide, and water as elements from whence chained arrows originate.

### 4.3. Interpersonal Metafunction

By and large, the students position themselves as knowledgeable in ecology (cf. 'fact oriented' in Løvland [64]). This is achieved by the choice of presenting facts through disciplinary languages, such as scientific terminology, chemical formulae, and disciplinary-specific symbols. Whereas naturalistically drawn elements depicting objects from the students' everyday experiences may lower the degree of scientific positioning, such choices may also indicate a fact-oriented position, as visual resources used in biology textbooks commonly contain such images. Some choices, such as the inclusion of comic book symbolism, point to an entertaining positioning (cf. [64]). By including fantasy strains, comic book symbolism, and explicit values, the students involve elements uncommon in the scientific discourse. However, through such choices, they connect ecology to their everyday life and give a view of themselves as creative and not fixed to scientific conventions. Schleppegrell [62] claims that for students to be successful in school, they need to develop an academic and abstract school language differing from their everyday language. However, students need support in approaching such academic language, which has been proven to be challenging for science teachers [68]. Thus, to increase students' interest, stronger connections to their everyday lives and interests should be made [66]. At the same time, the students need support in developing the scientific language. In addition, students' values and attitudes are visible in the texts, e.g., when expressing that meat-eating will lead to increased greenhouse gas levels, leading to 'the end of the world' (Figure 5). Such choices indicate a position of 'environmentally conscious citizen'.

### 4.4. Conclusions and Implications for Education

Much of earlier research claims various possibilities when students create drawings [32] and multimodal texts in science [38,69]. The findings of this study indicate that

students' representations and combinations of complex biological processes through multi-modal texts involve both possibilities and challenges. An example of a challenge concerning the use of semiotic resources is shown in the unclear use and meaning of arrows in the student texts. Interestingly, the students largely include humans in their depicted ecosystems, which can indicate an urge to emphasise the role of humans in ecological processes. In addition, most texts in some way indicate that photosynthesis is the basis for life on Earth. Through their textual choices, the students position themselves in various ways, often as knowledgeable in ecology (e.g., through disciplinary language), but they occasionally also take a humoristic position. Furthermore, the students include aspects uncommon in visual resources in ecology in their texts, such as symbols inspired by comic books, values, and the role of humans, thus relating ecosystems to their interests and everyday life.

The student texts with their various elements are in this study viewed as evidence of students' meaning-making in ecology. However, assignments given to students always direct the students toward making certain choices. In this case, they were directed towards using visual representations ('Show how . . . '). Even so, some student groups chose to use written language to a large extent. If, for example, the words 'show and describe' instead had been used in the assignment, the texts might have looked differently.

According to the Swedish curriculum, students are supposed to express science content in different ways, for example, through images and writing [3]. As this study has shown, this can be a challenge. Thus, teachers need to address challenges related to how the subject content is represented in teaching resources and also discuss students' choices when they create texts. In this study, the teachers mainly emphasised how 'nice' the drawings were, and they seldom asked any follow-up questions or discussed differences between texts in class. This is in accordance with earlier research showing that student texts in science classrooms rarely receive much attention after production [70,71]. A student text will never fully represent what students find challenging or obvious, as there is always room for interpretation. To get a better picture of students' views, teachers can gain from discussing different aspects of texts with the students, for example, what arrows might mean and what the students wanted to express when including them. In such discussions, apart from talking about text conventions, such as how arrows are commonly used, the focus will naturally also be on different biological processes and how these processes are connected (also see [56]). In that way, classroom discussions about students' texts can constitute a learning opportunity both concerning subject content and appropriate ways of expressing the content according to the discourse of the discipline (cf. 'disciplinary literacy', e.g., [72]). Such discussions can also act as a basis for formative teaching [73] (these ideas were further developed and discussed in [49]).

To be able to take a stand on ecological issues, e.g., on climate, energy supply, and resource utilisation, students must gain a holistic understanding of ecosystems. In ecology education, visual resources and teaching often focus on plants and animals while excluding humans [67]. Furthermore, in Western science tradition, humans are considered one of all animal species, while, in religious traditions, humans are viewed as separate from other animal species, dominating nature and thus superior [74]. When learning biology, an anthropocentric view is common, viewing humans as the center of the world [75]. Researchers argue that anthropocentrism can hinder students' views on human-nature relationships [75,76]. In this study, students included humans in their texts without being instructed to do so. The inclusion of humans could function as a basis for classroom discussions relating to the important aspect of humans to ecosystems.

The framework used in the present study can appear complex and extensive for teachers. However, as this study indicates, students need support when creating multimodal texts in ecology—as in other content areas. Experiences and insights concerning text production and the use of different resources in texts are crucial in other scientific fields as well as other school subjects. By using the framework used in the present study as a basis for text analysis and classroom discussions, many aspects otherwise easily overlooked can be brought to the fore. This, in turn, can give the teacher a better understanding of students'



views on ecology, increasing the opportunities for deeper, more scientific reasoning in the science classroom. In addition, as already mentioned, such discussion can support students' possibilities to develop their disciplinary literacy.

In the present study, emphasise has been on the product of students' group work when creating multimodal texts. A suggestion for future research is to include the process of group work in the analyses to get a wider picture of the students' meaning-making process. An upcoming article emphasises this process, for instance, regarding what different students contribute during the production of the multimodal text, both in terms of subject content and how the content is presented. The inclusion of humans in the student texts was an unexpected finding in the present study. An interesting topic for future research would be to investigate how humans and the role of humans are presented and can be included, in different scientific fields.

**Author Contributions:** Conceptualization, H.W., K.D. and S.W.; methodology, H.W., K.D. and S.W.; validation, H.W. and K.D.; formal analysis, H.W. and K.D.; investigation, H.W., K.D. and S.W.; data curation, H.W.; writing—original draft preparation, H.W. and K.D.; writing—review and editing, H.W., K.D. and S.W.; supervision, K.D. All authors have read and agreed to the published version of the manuscript.

**Funding:** This research received no external funding. We gratefully acknowledge Linnaeus University who made this study possible.

**Institutional Review Board Statement:** Ethical review and approval were waived for this study in line with the procedures when data was collected, based on the fact that it does not include any sensitive personal data. The study was conducted in accordance with ethical considerations stated by the Swedish Research Council regarding the requirements related to information, consent, anonymity, and the right to withdraw from the project.

**Informed Consent Statement:** Informed consent was obtained from all subjects involved in the study.

**Data Availability Statement:** The data are not publicly available due to the ethical requirements of the project, as per the confidentiality agreement established with the participants.

**Acknowledgments:** We thank the participating teachers and students for their contributions. We are also grateful for the valuable comments from Brita Johansson-Cederblad, Jonathan Clark, and the anonymous reviewers of a previous version of the article.

**Conflicts of Interest:** The authors declare no conflict of interest.

## Appendix A

Please see Figure A1 below.

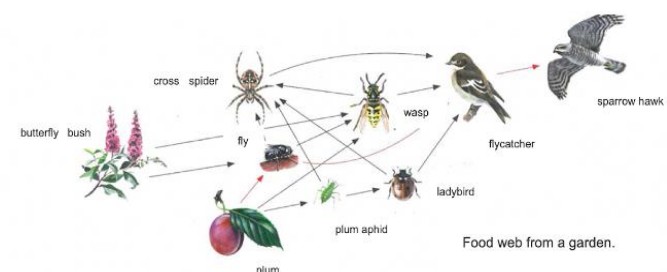

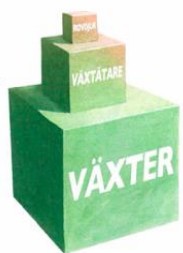

Show how the organisms in the food web get energy and materia.

   - how do animals get energy and matter?
   - how do plants get energy and matter?

Show what is needed in the food web in order for energy and matter not to run out.

Show what additional organisms that are needed for matter to be able to cycle.

The food pyramid shows that many plants are needed to support a small number of predators.

**Figure A1.** Assignment in ecology (translated from Swedish). Illustration from the textbook Henriksson, ref. [77] published by Gleerups Utbildning AB, the copyright for the illustration: Oskar Jonsson (reprinted with permission from Oskar Jonsson). The textbook uses the Swedish word 'näringspyramid' [nutrient pyramid], which has been translated to 'food pyramid' since the word 'näringsväv' has been translated to 'food web'.

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
