# Peer review of "Meaning-Making in Ecology Education: Analysis of Students’ Multimodal Texts"

_education, doi:10.3390/educsci13050443_

Round 1
Reviewer 1 Report
The authors analyse multimodal texts (posters?) created by students about the functions and relations of a food web in the context of biology education. The topic investigated in this study might be significant for the readers of Education Sciences. However, there are some issues that I recommend revising before publishing.
1.) The argumentation of the entire article is not entirely coherent. The objectives of the study are not entirely clear, and contradictory focuses seem to be set at various points. Although this is an exploratory study, it would be good to be more precise in the rationale.
Section 1.3 states: „The aim of this study is to understand students’ possibilities and challenges when making meaning of complex biological relationships in ecology, we have investigated how secondary students express complex biological processes through different resources in multimodal texts.“ The following two questions, in turn, refer on the one hand 'only' to the description of the addressed content by the students and on the other hand to how the students „position themselves […] regarding subject content“. It is not clear to me exactly what the study is supposed to be about. Is the focus on whether students correctly understand concepts of ecology (here: food web)? Or is it about showing the process in more detail, of how they come to understand concepts of ecology? Or is it more about whether including humans in food webs can support the understanding of ecological systems as a whole?
In Section 1.1 it is stated: „This is done without focusing on whether their expressed ideas are considered ‘right’ or ‘wrong’“. So, challenges in meaning making of compex biological relationsships are not meant to be with the content itself? But in section 4.1 the results are also discussed under the perspective of whether the content is correctly understood or whether the students' representations appropriately reflect the content. I am a little confused.
So, I would strongly reccomend making the whole argument more concise.
2.) In my opinion, the form of the assignement given to the students is a major factor influencing the possible results. Students creations are always also a reaction of the concrete requirements of the teacher and the material. Therefore, the role of the assignment in the interpretation should be discussed a bit more.
3.) The results are also strongly viewed very much from the perspective of the individual student. After all, the task was worked on in groups. Therefore, I would recommend interpreting the results from a social learning perspective as well. Especially the interesting analyses regarding the position of the students in the texts can be looked at in more depth there.
4.) This comment is more technical. The method of Systemic Functional Linguistics is well implemented and the analytical framework is convincing. However, how were the analyses intersubjectively validated? Was there a form of ‚double-coding‘, interpretation groups or some other methods?
5.) In my opinion, the most interesting results concern the interpersonal metafunction of the multimodal texts. If this could be more focused throughout the paper, it would make the paper more concise overall (see 1.) ).
Reviewer 2 Report
Thank you for the opportunity to read this manuscript. This study explored how secondary students use multimodal texts, including visual representations, symbols, and language, to represent and combine complex biological processes in the field of ecology. Results indicate challenges in the use and meaning of semiotic resources and the inclusion of uncommon aspects such as values and human roles. Implications for teaching include supporting students in representing content through text discussions. I have a good impression of this paper.
More specifically: This research provides valuable insights into how students use multimodal texts to represent complex biological processes, which is crucial for promoting a holistic understanding of ecology. The authors’ use of systemic functional linguistics to analyze students' multimodal texts is particularly innovative and insightful. I also welcome the authors’ attention to the challenges that students face in representing complex processes through various semiotic resources, as well as their creative solutions to these challenges.
The study's findings about the inclusion of uncommon aspects such as values and human roles in students' multimodal texts demonstrate a real-world relevance that is often lacking in academic research (unfortunately). Overall, this research makes an important contribution to the field of environmental education and offers valuable guidance for teachers and educators working with secondary students.
I wonder if the authors could further elaborate on the following questions, which were somehow addressed but need more specific examples/focus:
1. How did the researchers select the secondary students who participated in this study? Were there any specific criteria for inclusion?
2. What types of multimodal texts did the students create to represent and combine complex biological processes? Were there any notable differences in the types of texts produced by different students?
3. Did the study identify any specific strategies or approaches that students used to overcome the challenges of representing complex processes through multiple semiotic resources?
4. How might the findings of this study be applied in the development of ecology curricula or teaching resources for secondary students?
5. What are some potential avenues for future research in this area, and how might these expand on the insights provided by this study?
6. How do the findings of this study compare to previous research on multimodality in ecology education, if any?
7. Were there any unexpected or surprising results that emerged during the analysis of the students' multimodal texts?
8. How might the insights from this study be relevant to other fields of science education beyond ecology?
9. Were there any notable differences in the ways that students from different backgrounds or demographic groups represented and combined complex biological processes through multimodal texts?
10. What are some potential implications of this research for the development of students' scientific literacy more broadly?
Reviewer 3 Report
L. 23. "Ecology has a prominent role in curricula across all levels of schooling, from elementary to upper secondary school (e.g., [1-4])." No need to write "e.g." before literature references. Please include this comment throughout the article.
L. 47-56 in the given excerpt should still be added information that, for example, in the natural environment, many processes allow for a better understanding of individual processes in nature but it is not always possible to conduct classes in open, natural areas - see publications e.g.:
Hvenegaard, G. T. (2017). Visitors' perceived impacts of interpretation on knowledge, attitudes, and behavioral intentions at Miquelon Lake Provincial Park, Alberta, Canada. Tourism and Hospitality Research, 17(1), 79-90;
Korcz, N., Janeczko, E., & Kobylka, A. (2022). The Use of Simple Language in Informal Forest Education as a Key to the Correct Interpretation of Sustainable Forest Management-The Experience of Poland. International Journal of Environmental Research and Public Health, 19(9), 5493.
L. 440-453 is a redundant repetition of the article's assumptions and results in the discussion section. Please remove it.
Throughout the discussion, I missed specific references to the role of visualization, the role of images, graphics and simple language in education and in environmental education. Please consider including these issues.
L. 531-564 please specify in detail the final conclusions, what the authors got from their research
Round 2
Reviewer 1 Report
Thanks to the authors for the revision of the manuscript. In my opinion, it has gained in coherence and clarity.
Author Response
Thank's for the comment! We have gone through the text carefully and made some minor changes that hopefully contribute to the quality of the manuscript.